# Near quantitative synthesis of urea macrocycles enabled by bulky *N*-substituent

Yingfeng Yang[1,10], Hanze Ying[1,10], Zhixia Li[2,3], Jiang Wang[4], Yingying Chen[1], Binbin Luo[1], Danielle L. Gray [5], Andrew Ferguson [6], Qian Chen [1,3,7,9], Y. Z [2,3] & Jianjun Cheng [1,3,7,8,9 ✉]

Macrocycles are unique molecular structures extensively used in the design of catalysts, therapeutics and supramolecular assemblies. Among all reactions reported to date, systems that can produce macrocycles in high yield under high reaction concentrations are rare. Here we report the use of dynamic hindered urea bond (HUB) for the construction of urea macrocycles with very high efficiency. Mixing of equal molar diisocyanate and hindered diamine leads to formation of macrocycles with discrete structures in nearly quantitative yields under high concentration of reactants. The bulky *N-tert*-butyl plays key roles to facilitate the formation of macrocycles, providing not only the kinetic control due to the formation of the cyclization-promoting *cis* C = O/*tert*-butyl conformation, but also possibly the thermodynamic stabilization of macrocycles with weak association interactions. The bulky *N-tert*-butyl can be readily removed by acid to eliminate the dynamicity of HUB and stabilize the macrocycle structures.

[1] Department of Materials Science and Engineering, University of Illinois at Urbana-Champaign, Urbana, IL 61801, USA. [2] Department of Nuclear, Plasma, and Radiological Engineering, University of Illinois at Urbana-Champaign, Urbana, IL 61801, USA. [3] Beckman Institute for Advanced Science and Technology, University of Illinois at Urbana-Champaign, Urbana, IL 61801, USA. [4] Department of Chemistry, Rice University, Houston, TX 77005, USA. [5] George L. Clark X-Ray Facility & 3M Materials Laboratory, School of Chemical Sciences, University of Illinois at Urbana-Champaign, Urbana, IL 61801, USA. [6] Pritzker School of Molecular Engineering, University of Chicago, Chicago, IL 60637, USA. [7] Department of Chemistry, University of Illinois at Urbana-Champaign, Urbana, IL 61801, USA. [8] Department of Bioengineering, University of Illinois at Urbana-Champaign, Urbana, IL 61801, USA. [9] Materials Research Laboratory, University of Illinois at Urbana-Champaign, Urbana, IL 61801, USA. [10] These authors contributed equally: Yingfeng Yang, Hanze Ying. ✉email: jianjunc@illinois.edu

Macrocycles are unique structural units of very broad interest. Owing to their relatively rigid and defined scaffold, macrocycles are promising in biological sensing[1–3], ion transporting[4], drug discovery[5–7] and molecular sieving[8]. Building on their propensity for self-assembly, macrocycles can also serve as precursors to mechanically interlocked structures[9,10] and molecular machinery[11], and play an important role in catalysis[12] by providing nano-confinement[13]. The selective binding and associated host–guest interaction further dictate their usage in environmental pollutant removal[14] or drug delivery[15]. Despite all these widespread applications, synthesis of macrocycles remains challenging, in particular under high concentration in large scale.

Traditionally, macrocycles are synthesized through kinetic control in very dilute concentrations, which suffers from low yields and tedious purification processes[16]. With the advent of dynamic covalent chemistry (DCC), syntheses of macrocycles with improved yields through thermodynamic control have been reported[17]. Aided by the 'error checking' and 'self-correction' features enabled by reversible reactions, DCCs like boronic ester[18], imine[19], alkene[20] and alkyne[21] metathesis have been demonstrated for macrocycle construction with moderate yields. Since the linkage between the building blocks can greatly impact the physical and chemical properties of the material, new, viable DCCs are desired for the construction of macrocycles, in particular those of intrinsic hydrogen bonding capabilities which are of huge implications in molecular recognitions, self-assembly, molecular machinery and catalysis.

Cyclic peptides have been important hydrogen bonding-capable moiety in self-assembly[22] and are key structures of numerous natural pharmacophores[23]. Synthesis of cyclic peptides in high yields, however, is quite challenging. Synthesis of macrocyclic amides, which share the identical backbone functional group as cyclic peptides, have been extensively reported with structurally rigid aromatic precursors with pre-designed bond angles or by employing intramolecular hydrogen bonding to pre-organize the building blocks[24]. However, there have been no viable DCC reported so far for efficient macrocyclic amide synthesis. As a structural surrogate, urea shares desirable features with the amide group, such as rigidity, planarity, polarity and hydrogen bonding capacity. There have been numerous reports on binding, recognition, catalysis or assembly of acyclic oligoureas[25–27]. Synthesis and applications of urea macrocycles have also been explored. For instance, Shimizu reported bisurea macrocycles as nanocontainer to facilitate photodimerization, selective cycloaddition and to stabilize radicals[28–30]. Gong pioneered an efficient approach for one-pot formation of aromatic tetraurea macrocycles via specific oligomeric structures preorganized by their intrinsic intramolecular hydrogen bonds[31]. Gale reported a tetraurea macrocycle that can bind chloride in aqueous solution[3]. However, these reports typically involve synthesis of urea macrocycles in mediocre yields or with specific building blocks, and some even require stringent reaction condition and reagents. Here, we report a simple, robust method for constructing urea macrocycles with hindered urea bond (HUB) associated dynamic chemistry. A variety of hindered urea macrocycles (HUMs) are synthesized in near quantitative yields under high concentrations, with inexpensive and widely available diisocyanates and bulky diamines. The unusual high efficiency is attributed to the steric conformation lock and weak association interactions mediated by the bulky group as well as the 'self-correction' property enabled by the dynamic bond. The bulky tert-butyl (t-Bu) group can be readily removed by acid to stabilize the macrocycle structure, facilitating their applications in self-assembly, binding and molecular recognition and therapeutics.

## Results

**Rationale for the efficient construction of HUMs.** In an amide with chemical structure of $R^1C(O)NHR^2$, the $R^1$ and $R^2$ moieties should be 'trans' to each other (Fig. 1a) because of the coplanar resonance structure of the amide bond (C(O)–NH) as the otherwise 'cis' conformation of these two moieties would result in significant steric hinderance and is therefore thermodynamically disfavored. Same 'trans' bond structure should also hold true in the case of an urea with chemical structure of $R^1N^1HC(O)N^2HR^2$, the close analog of amide, with $R^1N^1H$ moiety 'trans' to the $R^2$ moiety in the coplanar resonance structure of the urea bond ($N^1C(O)$–$N^2H$) (Fig. 1b). These have been the basis forming the zigzag molecular structures of the polyamide or polyurea backbones. However, such $R^1/R^2$ 'trans' bond structures of the amide $R^1C(O)NHR^2$ may no longer hold true when the hydrogen of the N–H was replaced by a substituent much bulkier than $R^2$. It has been reported that when the N–H was changed to N-t-Bu in arylopeptoids or α-peptoids, the amide bonds adopted the $R^1/R^2$ cis conformation exclusively (Fig. 1c)[32]. We previously reported HUB as a new DCC tool[33], which can be synthesized from isocyanates and hindered amines, and can dissociate back to the starting compounds (Fig. 1e). As the t-Bu has been predominantly used in our design of HUB, we postulate that replacing the hydrogen of $N^2H$-$R^2$ in the urea by t-Bu (bulkier than $R^2$) would result in $R^1$-$N^1H$ 'cis' to $R^2$ (i.e., bulky t-Bu N-substitute 'cis' to C(O)) (Fig. 1d). Throughout this paper, we use 'cis-urea' to define such bond structure with bulky N-substitute cis to C(O), as shown in Fig. 1d. If such 'cis-urea preference' holds true in a HUB containing bulky t-Bu group, we envision that the [1:1] adduct of a diisocyanate and a hindered diamine would result in a hindered polyurea with 'bulky urea-turn' ('cis' structure of $R^1NH$ and $R^2$, Fig. 1f) in their backbone instead of forming a zigzag urea backbone. For a typically [1:1] adduct of a diisocyanate and a diamine with perfect zigzag urea chain without bulky groups, since the backbone units all prefer to stay trans to each other, macrocyclization would be difficult in the linear extended backbone structure and can only occur in highly dilute solution. While for hindered urea chains with 'bulky urea-turn' in each HUB, such hindered polyurea should easily undergo macrocyclization with several bond rotations (Fig. 1g). This resembles to some extent the so-called 'gem-dimethyl effect' where large substituents favor ring-closure and intramolecular reactions[34,35]. Thus, we envision that the properly selected diisocyanate and hindered diamine adducts adopting significant 'bulky urea-turn' conformation would enable efficient construction of urea macrocycles.

**Quantitative synthesis of the hindered urea macrocycle (HUM1).** In a preliminary trial, two commercially available building blocks, methylene diphenyl diisocyanate (**N1**) and N, N'-Di-tert-butylethylenediamine (**A1**), were mixed (Fig. 2a) in 1:1 molar ratio and maintained at 60 °C. The system initially gave mixtures of multiple species. Interestingly, after incubating for 2 h, the originally messy and multimodal peaks transformed to one set of sharp resonances in $^1H$ NMR (Fig. 2b), implying the formation of possibility a low molecular weight residue with well-defined structure in very high purity. The residue was characterized by multiple methods without any purification (Supplementary Fig. 2). Evidences pointed to the anticipated formation of a dimeric **HUM1** in quantitative yield with the unnecessity of any purification, by simply mixing equal molar **N1** and **A1**. The structure of **HUM1** was further confirmed by the single-crystal X-ray diffraction (XRD) analysis (Fig. 2c).

To better understand the formation process of **HUM1**, the reaction was conducted at room temperature in high concentration

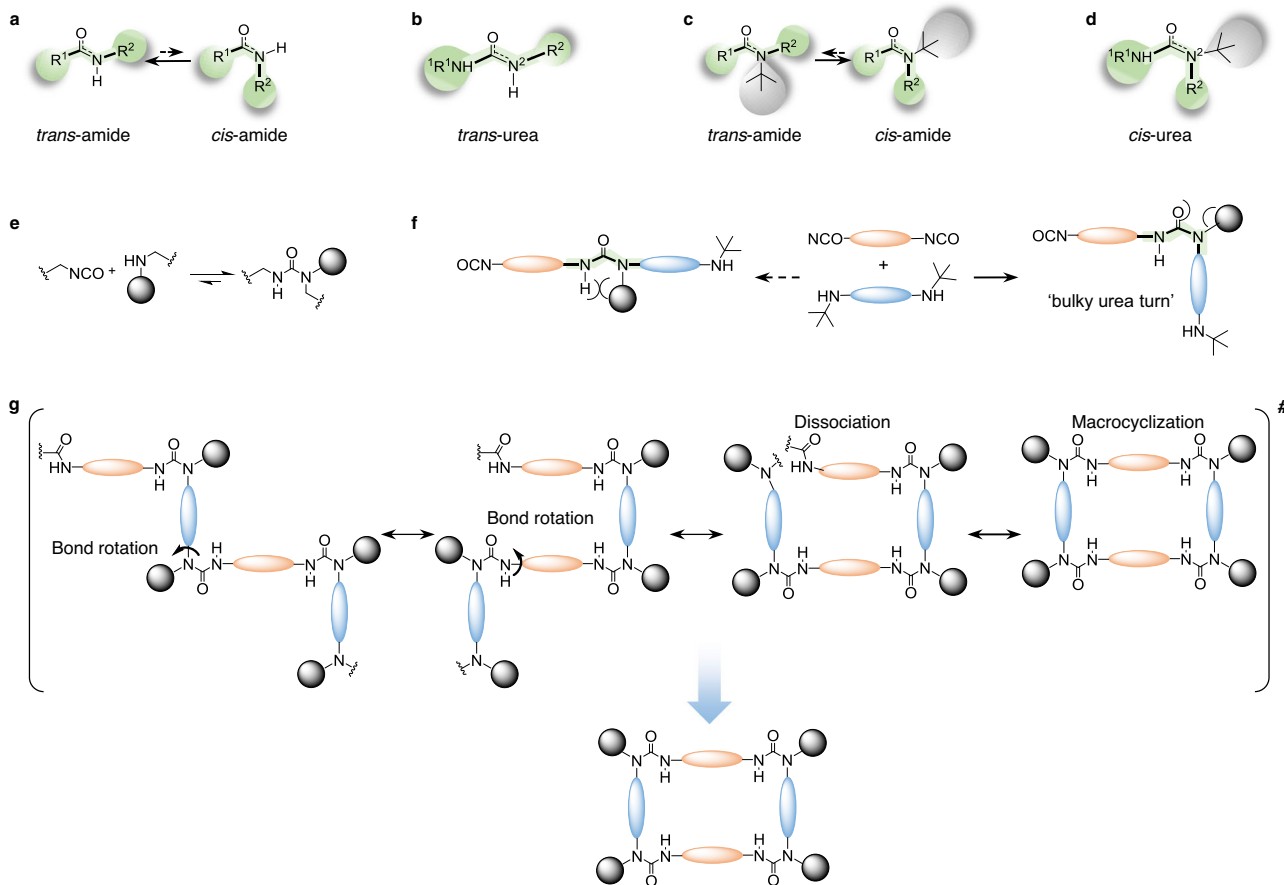

**Fig. 1 Rationale for the efficient construction of HUMs. a** *Trans* conformation preference of typical unhindered amide. **b** *Trans* conformation preference of typical unhindered *N, N'*-di-substituted urea. **c** *Cis* conformation preference of amide with bulky *N*-substituent. **d** Proposed *cis* conformation of ureas with bulky *N*-substituents. **e** Schematic illustration of the hindered urea bond (HUB). The bulky substituent broke the bond co-planarity, rendering HUB dynamic. **f** Schematic illustration for 'bulky urea-turn' and hindered urea chains. **g** Schematic illustration for the *cis*-urea facilitated macrocyclization. If the bulky group induced '*cis*-urea preference' holds true, the hindered urea chain will bend and can easily undergo macrocyclization with several bond rotations.

and monitored by gel permeation chromatography. A gradual transition from higher molecular weight multimodal peaks to one sharp peak with the calculated molecular weight consistent with **HUM1** was observed (Fig. 2d), which suggests that the final thermodynamic equilibration has been reached. Furthermore, **HUM1** can be quantitatively obtained within a wide range of concentration, from 1 to 500 mM (entries 1–4, Table 1) at any temperature ranging from 20 to 75 °C (entries 5–7, Table 1). Reactions of equal molar **N1** and **A1** in different aprotic solvents, such as chloroform, THF, ethyl acetate, dimethyl formamide and dimethyl sulfoxide, all result in the formation of pure **HUM1** in quantitative yields, with only differences in their time of reaching the equilibrium to form **HUM1**.

**HUB as macrocycle enabling structural motif for the synthesis of other HUMs.** The exceptionally high efficiency of **HUM1** synthesis prompts us to explore if the reaction is universal to other substrates. Four commercially available diisocyanates Nx (x = 1–4) were chosen, as well as five hindered diamines Ay (y = 1–5) which are either commercially available or can be facilely synthesized from *t*-butylamine and the corresponding halides (Table 1). The macrocycle formation efficiencies were monitored by [1]H NMR and the final products were further confirmed by [13]C NMR and matrix-assisted laser desorption/ionization-time of flight (MALDI-TOF). Most of the combinations in the library gave HUMs as the predominant species within short period of

time (entries 1 and 8–22, Table 1, Supplementary Figs. 9–25). Among the 20 combinations of NxAy, 13 of them gave quantitative yield of macrocycles. While most combinations gave [NxAy]$_2$ dimeric macrocycles as shown in Table 1, [NxAy]$_1$ type of monomeric macrocycles were formed in some systems involving **N2**, presumably because the relatively flexible benzyl linker of **N2** can effectively release the ring strain for the monomeric macrocycle species. The combinations of **N2** with **A2**, **A3** and **A5** even reached exclusive macrocycle product at room temperature within 3 min after mixing (entries 9, 10 and 12), much faster than the dynamic exchange of the reversible bond[33,36]. It is thus evident that the HUB is a remarkable macrocycle enabling structural motif, given the unusually high yield of macrocycles from the off-the-shelf, readily available building blocks.

The macrocycle formation was also tested in a three-component system where two different diisocyanates **N3** and **N4** were mixed with one diamine **A2** (entry 23, Table 1, Supplementary Fig. 28). A self-sorting phenomenon was observed, in which HUMs **[N3A2]$_2$** and **[N4A2]$_2$** formed, respectively, with no hybrid products detected. The self-sorting presumably results from the dynamicity of the system which allows it to collapse to its thermodynamically most stable state, as otherwise mismatch of the building blocks will possibly incur ring strain and thus increase total energy of the system. Postmodification of macrocyclic scaffold offers a higher level of manipulation. We demonstrated that the presence of the pyridyl and the pendant alkynyl groups in monomers **A4** and **A5** did not

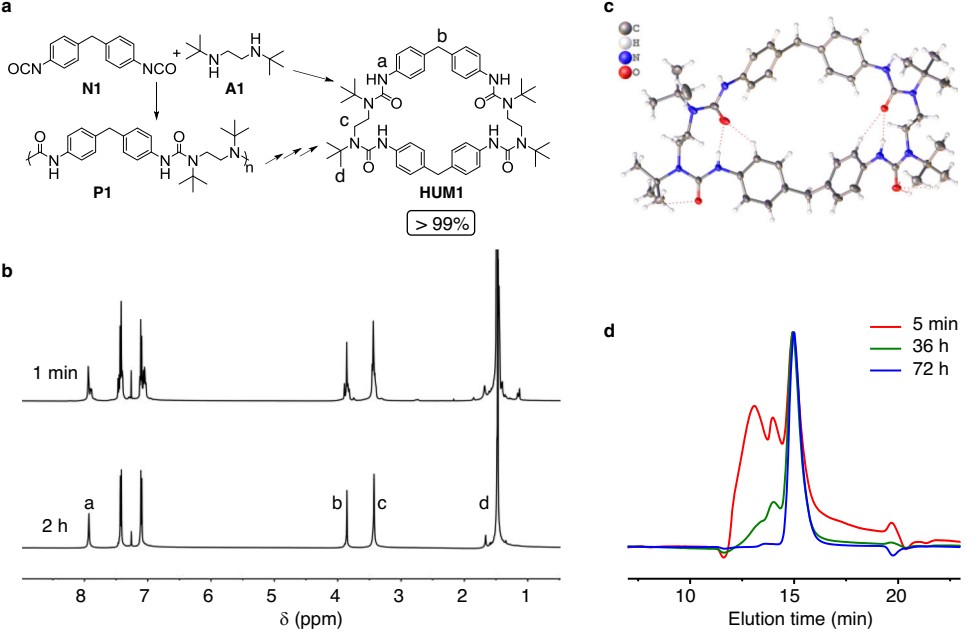

**Fig. 2 Quantitative synthesis of the hindered urea macrocycle HUM1. a** Structures of **N1**, **A1**, their corresponding oligomer **P1** and the dimeric macrocycle **HUM1**. **b** $^1$H NMR of the 1:1 mixture of **N1** and **A1** (50 mM) upon mixing and after 2 h incubation at 60 °C in CDCl$_3$. The sharp and clean peaks demonstrated near quantitative yield of **HUM1**. **c** Single-crystal structure of **HUM1**. Color code: gray, C; white, H; blue, N; red, O. **d** Normalized GPC trace showing the formation process of **HUM1** (200 mM in chloroform, 20 °C). After incubation, broad and multimodal peaks (red trace) gradually transformed into a sharp peak (blue trace), demonstrating the equilibration process and the near quantitative yield of **HUM1**.

interfere with the macrocycle formation process, implying the potential for further functionalization and structural diversity extension in future designs. What's more interesting, when the *t*-Bu group in **A2** was replaced with even bulkier ones (**A6** and **A7**), similar macrocycles formation behaviors were shown (entries 24–25, Supplementary Figs. 26 and 27), further substantiating the versatility of this strategy.

Among various combinations with exclusive macrocycle formations, two types of processes were observed: the mixings of diisocyanate and diamine either gave high-purity macrocycle immediately (entries 9, 10 and 12, Table 1) as kinetically favored products or gave mixture which later turned to macrocycles exclusively as thermodynamically favored products. These two types were further investigated to understand the underlying mechanisms.

**Validation of *t*-Bu induced 'cis-urea preference'.** The observation of kinetically controlled macrocycle formation process is consistent with our initial hypothesis (Fig. 1g). To validate the relationships between the 'cis-urea' conformation and size of substituents, the kinetically favored macrocycle **HUM2** ([N2A2]$_1$) was selected as well as a series of acyclic model compounds (**MC1** and **MC1′**, which differ only in one substituent) (Fig. 3). Their conformations were studied by Nuclear Overhauser Effect Spectroscopy (NOESY). While substituted urea usually adopts various conformations due to the relatively low-rotational barriers of urea C-N bond[37,38], both **HUM2** (Fig. 3a) and the *t*-Bu substituted **MC1** (Fig. 3b) adopted a relatively fixed *cis* conformation, with the N–H hydrogen (labeled as 'a') in close proximity only to the less bulky benzyl/methyl hydrogens (labeled as 'b') but not to the *t*-Bu hydrogens (labeled as 'c'). On the contrary, if the *t*-Bu group was changed to an ethyl group as in **MC1′**, the NH became spatially close to both the methyl and ethyl group (Fig. 3c), showing the C(O)–N(Me)Et amide bond can rotate more freely with both conformations coexisting. Density functional theory (DFT) calculations revealed more than 4.2 kcal/mol (~7 kT) energy differences

between *cis/trans* conformations of the *t*-Bu substituted model compound **MC2**, which is much larger than its *i*-Pr substituted counterpart **MC2′** (~2 kT) (Fig. 3d). Similar NOE and DFT calculation results were obtained for different sets of model compounds representing aromatic/aliphatic ureas (Supplementary Figs. 35–38, Supplementary Table 1, Supplementary Information 3.3.2, 3.3.3). These results clearly demonstrated the 'cis-urea preference' which is induced by the bulky *t*-Bu substituent.

We next elucidated how the 'cis-urea preference' can facilitate macrocycle formation by using the linear analog **MC4** (the [1:1] adduct of **N2** and **A2** with reactive chain ends, upper left inset, Fig. 3f, Supplementary Fig. 39). Twelve different conformations were generated by rotating around the C(O)–N(*t*Bu) bond of the adduct, which converged into five different metastable conformations after local energy minimization. The DFT calculations on the five different conformers of **MC4** revealed the general trend that with increased degree of folding (i.e., smaller θ1 or shorter chain-end distance d), the free energy decreases, substantiating that the conformer with the shortest reactive chain-end distance predominates. It is also observed that the most stable conformers adopt 'cis-urea' conformations (lower right inset, Fig. 3f), elucidating the correlations between 'cis-urea preference' and proximity of reactive chain ends, which contributes to the efficient macrocycle formation.

While only **HUM2** was discussed here, kinetic effects should also play a role in dimeric macrocycle systems. This argument is supported by the cis-urea conformation of several macrocycles and linear analogs shown by the single-crystal structures (Supplementary Figs. 54–57, Supplementary Tables 4–7), and the fact that most high-yielding HUMs systems underwent a burst formation with more than 40% yields of the target macrocycles in less than 3 min, which was much higher than the statistic yields at such concentration (Supplementary Fig. 12, Supplementary Information 3.2.4). Thus, the kinetic aspect of the 'cis-urea preference' facilitated efficient HUM formation can be rationalized from two aspects: conformational flexibility and effective molarity. On one

**Table 1 Substrate universality of HUB based macrocycles.**

| Entry | Isocyanates | Amines | HUMs | Conc. (mM) | T (°C) | t (h) | Yields (%) |
|---|---|---|---|---|---|---|---|
| 1 | N1 | A1 | [N1A1]$_2$ | 50 | 60 | ~1 | >95 |
| 2 | N1 | A1 | [N1A1]$_2$ | 1 | 60 | ~0.5 | >95 |
| 3 | N1 | A1 | [N1A1]$_2$ | 200 | 60 | ~1.5 | >95 |
| 4 | N1 | A1 | [N1A1]$_2$ | 500 | 60 | ~2 | >95[a] |
| 5 | N1 | A1 | [N1A1]$_2$ | 50 | 20 | ~48 | >95 |
| 6 | N1 | A1 | [N1A1]$_2$ | 50 | 37 | ~12 | >95 |
| 7 | N1 | A1 | [N1A1]$_2$ | 50 | 75 | ~0.5 | >95 |
| 8 | N2 | A1 | [N2A1]$_2$ | 50 | 60 | ~20 | 61 |
| 9 | N2 | A2 | [N2A2]$_1$ | 50 | 20 | <3 min | >95 |
| 10 | N2 | A3 | [N2A3]$_1$ | 50 | 20 | <3 min | >95 |
| 11 | N2 | A4 | [N2A4]$_1$ | 50 | 20 | <3 min | >95 |
| 12 | N2 | A5 | [N2A5]$_1$ | 50 | 20 | <3 min | 91 |
| 13 | N3 | A1 | [N3A1]$_2$ | 50 | 60 | ~3 | 81 |
| 14 | N3 | A2 | [N3A2]$_2$ | 50 | 60 | ~2 | >95 |
| 15 | N3 | A3 | [N3A3]$_2$ | 50 | 60 | ~1 | >95 |
| 16 | N3 | A4 | [N3A4]$_2$ | 50 | 60 | ~3 | >95 |
| 17 | N3 | A5 | [N3A5]$_2$ | 50 | 60 | ~2 | >95 |
| 18 | N4 | A1 | [N4A1]$_2$ | 50 | 60 | ~2 | 68 |
| 19 | N4 | A2 | [N4A2]$_2$ | 50 | 60 | ~2.5 | >95 |
| 20 | N4 | A3 | [N4A3]$_2$ | 50 | 60 | ~1 | >95 |
| 21 | N4 | A4 | [N4A4]$_2$ | 50 | 60 | ~3 | >95 |
| 22 | N4 | A5 | [N4A5]$_2$ | 50 | 60 | ~2 | >95 |
| 23 | N3 + N4 | A2 | [N3A2]$_2$; [N4A2]$_2$ | 25; 25; 50 | 60 | / | >95 |
| 24 | N3 | A6 | [N3A6]$_2$ | 50 | 60 | / | >95 |
| 25 | N3 | A7 | [N3A7]$_2$ | 50 | 60 | / | >95 |

Upper: Schematic illustration for the formation of monomeric macrocycles [NxAy]$_1$ or dimeric macrocycles [NxAy]$_2$ from the combination of diisocyanates Nx and hindered diamines Ay, and the corresponding structures of Nx and Ay (x = 1–4, y = 1–7). Orange: diisocyanates; blue: hindered diamines. Bottom: Yields of HUMs for different representative combinations under different conditions. Yields were determined by $^1$H NMR.
[a]Product precipitated out as white crystals because of limited solubility of the macrocycle.

hand, the 'cis-urea preference' would reduce the conformational flexibility of the urea chain and thus, the minimum of the energy landscape can be reached faster[39]. On the other hand, the folded structure will position the reactive chain ends close to each other, thus increasing the effective molarity for the ring-closing step. In cases where the building blocks are suitably angled with chain ends in proximity, monomeric macrocycles can be quantitatively formed upon mixing, such as **HUM2**. In cases where building blocks are

structurally more rigid, dimeric macrocycle structures that can accommodate four 'bulky urea turns' (Fig. 1f) are favored and can be formed with several bond rotations. Furthermore, in a situation where one urea bond of the macrocycle was opened up, the preorganized intermediate would provide a high effective molarity driving rapid ring-closure before another urea bond opens up, which in turn contributes to the higher macrocycle stability against exchange reactions. To prove this, **HUM2** and its linear model

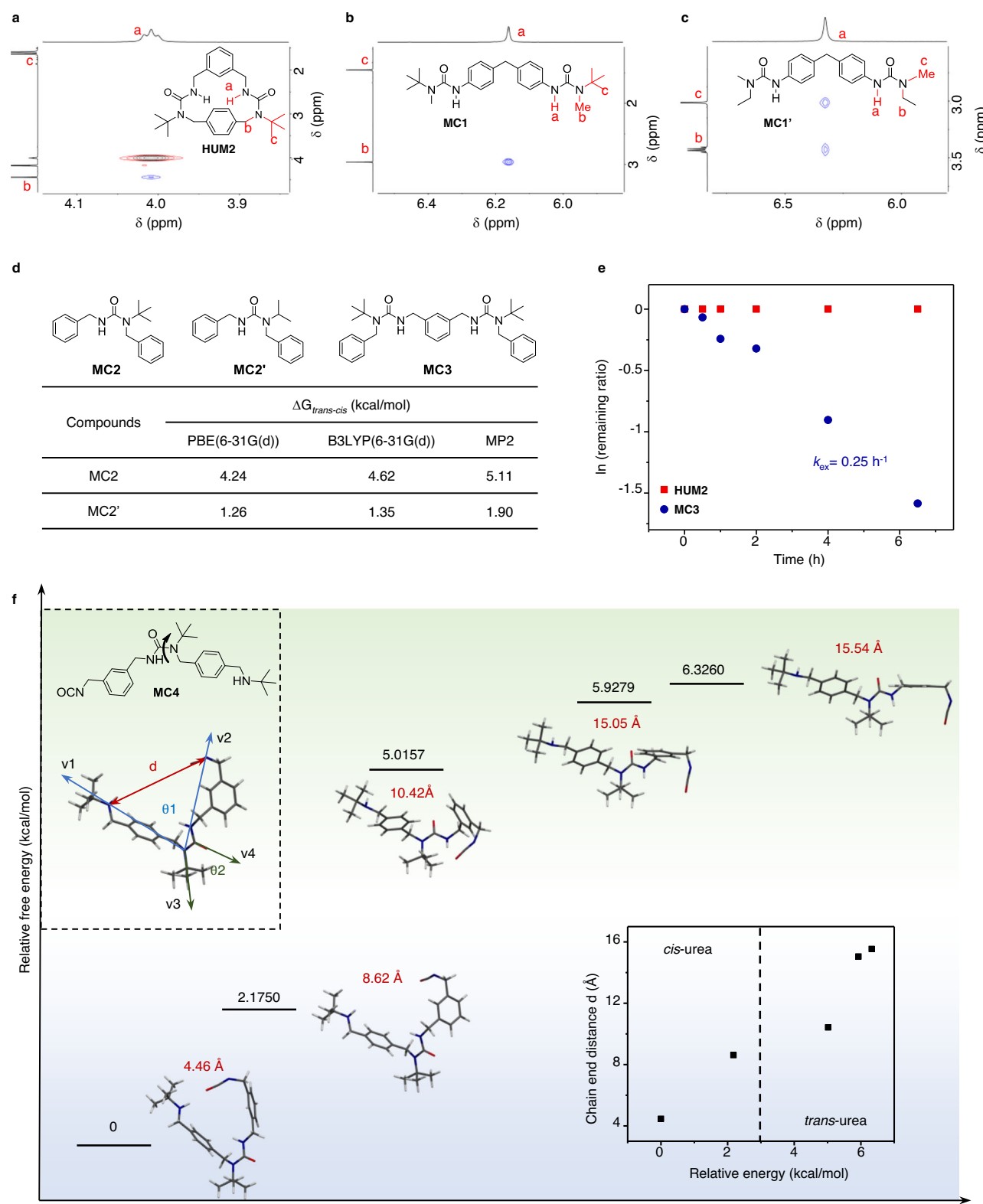

| Compounds | $\Delta G_{trans\text{-}cis}$ (kcal/mol) | | |
|---|---|---|---|
| | PBE(6-31G(d)) | B3LYP(6-31G(d)) | MP2 |
| MC2 | 4.24 | 4.62 | 5.11 |
| MC2' | 1.26 | 1.35 | 1.90 |

$k_{ex}$= 0.25 h$^{-1}$

compounds **MC3** were exchanged with butylisocyanate (Fig. 3e). The linear analog underwent fast exchange reactions while **HUM2** remained unchanged even after prolonged incubation of over 50 h.

**Role of *t*-Bu group in thermodynamic stabilization**. While '*cis*-urea preference' well explained the fast HUMs formation, the

phenomenon of equilibrium was beyond the explanation. In the first example of **HUM1** we discussed, a mixture of species with various sizes were formed first, which was later transformed to an exclusive macrocyclic compound during a relatively long period of time compared to kinetically favored macrocycles. It was a typical thermodynamic equilibrium process, which indicated that **HUM1** has reached a locally minimum energy state. In one of the

**Fig. 3 Validation of the *t*-Bu induced '*cis*-urea preference'.** Partial NOESY spectrum of **a** HUM2 **b** MC1 and **c** MC1'. HUM2 and MC1 showed relatively restrained *cis* conformation while MC1' had a relatively flexible conformation with free rotation. **d** Upper: structures of the model compounds used in the DFT calculations (MC2 and MC2') and that used in the exchange kinetic studies (MC3). Lower: computed free energy differences between *cis* and *trans* conformations of MC2 and MC2'. **e** Exchange kinetics of HUM2 ([N2A2]₁, red square) and its linear model compounds MC3 (blue dot) with 20 equivalents of butylisocyanate in CDCl₃ at 55 °C. **f** Illustration showing the structures of the five different conformers of MC4, their respective degree of folding θ1, relative-free energies and chain-end distance d. Conformations were obtained after local energy minimization. DFT calculations shown here were based on the B3LYP/6-31G(d) levels of theory. Upper left inset: structure of the adduct MC4 and the corresponding calculated parameters. d: distance between the N of the free amine and C of the free isocyanate. θ1: angle between vectors v1 and v2; θ2: angle between vectors v3 and v4. Lower right inset: relationship between the free chain-end distance and the relative-free energy.

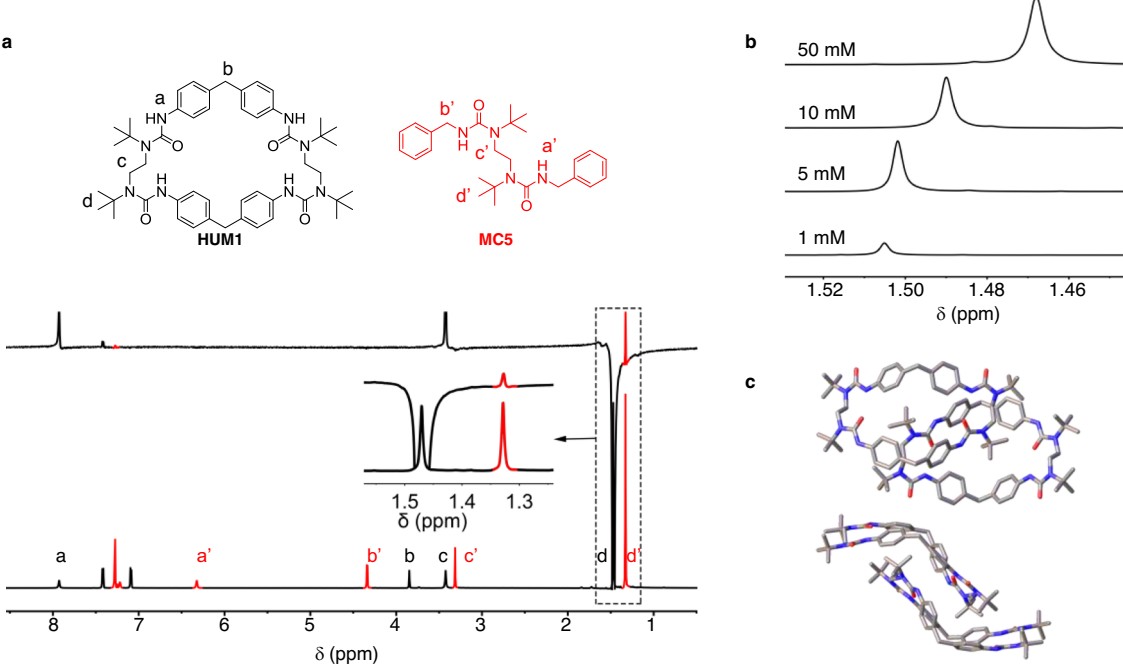

**Fig. 4 Role of *t*-Bu group in thermodynamic stabilization. a** Upper: structure of **HUM1** and the acyclic model compound **MC5** used in the NOE experiment. Lower: NOESY showing the spatial proximity between **HUM1** (black) and **MC5** (red) protons (750 MHz, CDCl₃, **HUM1** 25 mM, **MC5** 50 mM); when irradiating the *t*-Bu group in **HUM1**, **MC5** showed NOE peaks. **b** Concentration-dependent NMR of **HUM1**; peak for *t*-Bu group shifted upfield while other peaks showed no clear change. **c** Top and side view of **HUM1** from the single-crystal XRD; *t*-Bu group sits right inside the pocket. Color code: gray, C; blue, N; red, O.

explorative studies, we performed a similar reaction with **N1** and a new diamine with only the *t*-Bu group changed to *i*-Pr (**A1'**). In this case, a mixture of oligomeric molecules was obtained even with prolonged incubation (Supplementary Fig. 42). The mixture was proven to reach chemical equilibrium after 15 days without any trend towards an exclusive macrocyclic product. It implied some unique driving forces *t*-Bu group may provide to the formation of macrocycle that less bulky groups do not have. Then, a concentration-dependent NMR study was performed for **HUM1** solutions in CDCl₃ from 1 to 50 mM. Interestingly, the proton signal from the *t*-Bu group showed a clear trend of upfield shift with the increase of concentration (Fig. 4b, Supplementary Fig. 45), based on which the dimeric association constant of **HUM1** in CDCl₃ was determined to be around 5.5 M⁻¹ (Supplementary Fig. 46). On the contrary, the urea proton showed no clear change to the concentration, indicating no association through hydrogen bonds. As a control, no such phenomenon was observed for its acyclic analog **MC9** (Supplementary Fig. 47). It implies the existence of an intermolecular interaction involving *t*-Bu group in macrocycles, and it was further supported by the NOESY study. As it is difficult to tell apart the 'through bond' or 'through space' coupling in the same molecule, the acyclic model compound **MC5** was added to the **HUM1** solution and the

mixture was characterized by NOESY (Fig. 4a). When the proton on *t*-Bu of **HUM1** (peak d) was irradiated, the signals from *t*-Bu protons of **MC5** showed up (peak d') as well as ones from the aromatic region of **MC5**. This demonstrated that the *t*-Bu groups of **HUM1** and **MC5** are in close proximity in space, supporting the argument of existence of intermolecular interaction involving *t*-Bu group.

The single-crystal XRD showed how *t*-Bu group may interact with the macrocycle in the solution state. As shown in Fig. 4c, two adjacent macrocycles both have one of their *t*-Bu groups pointing towards the C-shaped cavity of each other, forming a 'host-guest' pair. The corresponding *t*-Bu—cavity centroid distance is ~0.272 nm, suggesting that the macrocycle **HUM1** may be stabilized by the presence of energetically favorable *t*-Bu—macrocycle interactions. As a further support, atomistic level molecular simulation was performed in [N1 + A1] system to calculate the average monomeric non-bonding energy in macrocycles with various sizes n (denoted as nmer), either with or without intermacrocycle interactions considered (Supplementary Fig. 48a). The results suggest that the interactions between macrocycles stabilize the 2mers and drive it to be the much more favored species. Dimerization potential of mean force (PMF) calculations showed that 2mer had a higher stabilization free energy (4.0 kT)

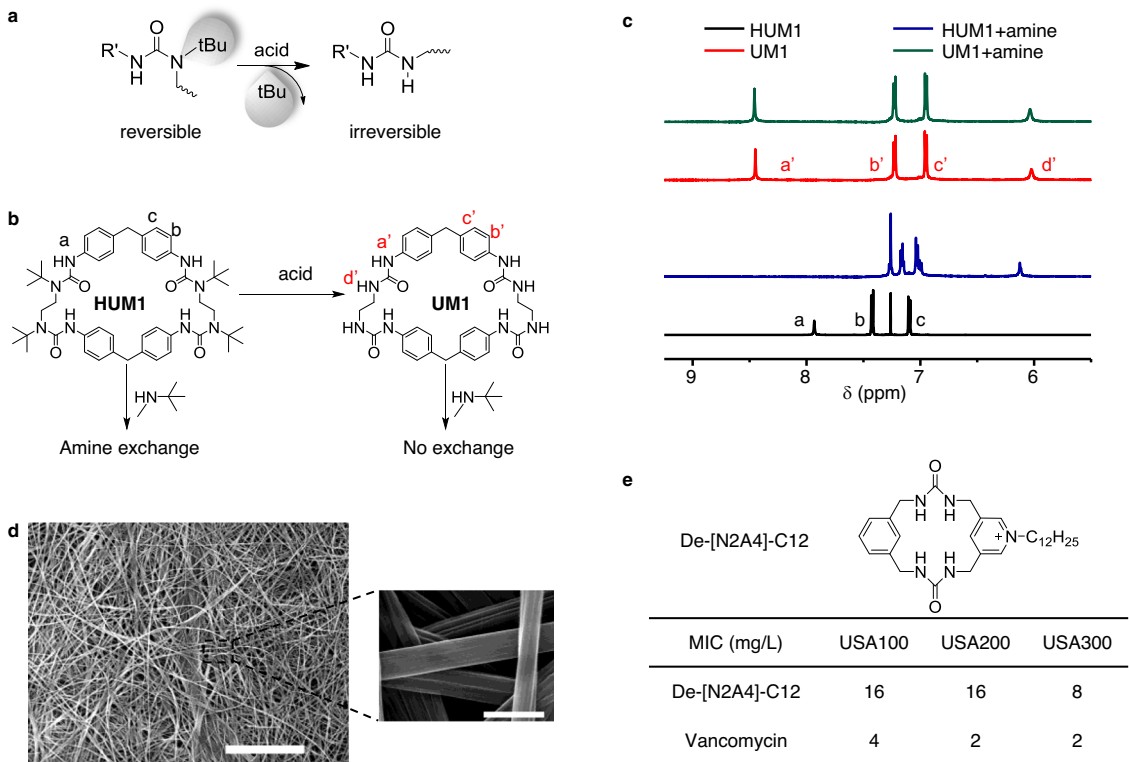

**Fig. 5 Stabilization of the HUMs through de-*tert*-butylation and the potential applications. a** Schematic illustration of the acid-assisted de-*tert*-butylation process. **b** De-*tert*-butylation of **HUM1** gave a stable urea macrocycle **UM1**, which remained unchanged with the amine exchange reaction. **c** ¹H NMR of **HUM1** (black, in CDCl₃) and **UM1** (red, in DMSO-d⁶). After adding amine (10 eq) and incubated for 1 h, **HUM1** was degraded (blue, in CDCl₃), while **UM1** still remained intact (green, in DMSO-d⁶). **d** SEM image of the self-assembled fiber structure of **UM1**. Scale bar: 20 μm (zoomed: 500 nm). **e** MIC (minimum inhibitory concentration) of the cationic charged urea macrocycle De-[N2A4]-C12 and the antibiotic vancomycin against three different Methicillin-resistant Staphylococcus aureus bacterial strains.

than other-sized macrocycles (Supplementary Fig. 48c). Furthermore, a 'pocket effect' was clearly observed in the 2mer system, with *t*-Bu group from one macrocycle sitting in the cavity of another one (Supplementary Fig. 48e), which was in consistence with the structure in the solid state.

**Stabilization of the macrocycles through de-*tert*-butylation reaction**. Macrocycles constructed from DCCs often suffer from stability issue because the reversible property of dynamic bonds still exist in ring molecules. We attempted to transform dynamic HUB structures to a regular urea to remove the dynamicity and stabilize the macrocycle. Given the structural similarity of HUB to *t*-Bu ester which can be readily cleaved by acid[40], we attempted to use acid to remove the *t*-Bu group from HUB (Fig. 5a). **HUM1** was treated with trifluoracetic acid for 5 min. As expected, the stable urea macrocycle **UM1** was obtained in nearly quantitative yield from the de-*tert*-butylation of **HUM1** (Fig. 5c). To assess its stability, we treated **UM1** with excessive amount of *N-tert*-butylmethylamine (Fig. 5b). **UM1** stayed unchanged with amine exchange reaction, substantiating the complete elimination of the dynamicity. In comparison, **HUM1** was quickly degraded via rapid amine exchange reaction with *N-tert*-butylmethylamine (Fig. 5c).

Apart from the increased stability, de-*tert*-butylated urea macrocycles show much stronger tendency to self-assemble. When **UM1** was dissolved in DMSO and set aside for 2 days, it self-assembled into fibers with 200~300 nm in width and greater than 200 μm in length, presumably assisted by the intermolecular interactions, such as π–π interaction, and the availability of NH for hydrogen bonding after de-*tert*-butylation (Fig. 5d). As expected,

urea macrocycles can also be used on molecular recognition. For instance, **UM1** was found to be a potent receptor for several different anions (Supplementary Fig. 51). The binding constant between **UM1** and phosphates, for example, can reach $10^4$ M⁻¹ even in competitive solvents like DMSO (Fig. S34).

Inspired by the structural similarity between urea macrocycles and cyclic peptides and the notion that cyclic structures usually have improved membrane activity compared with their linear analogs, we further explored the biological activities of urea macrocycles as antimicrobial agents. **[N2A4]₁** was selected because of its rigid structure and the ease of postmodification via its pyridinyl moiety. Through alkylation, the charged urea macrocycle with C12 alkyl tail showed high antimicrobial activities against three different strains of methicillin-resistant staphylococcus aureus (MRSA) (Fig. 5e), with the minimum inhibitory concentration comparable to Vancomycin, a clinically widely used antibiotics for MRSA treatment. Following these preliminary studies, explorations of other applications of urea macrocycles are under way.

## Discussion

In this research, urea macrocycles are synthesized efficiently through DCC under ambient conditions from readily available starting materials. The strategy shows high versatility to different substrates, reaction conditions and bulky substituents, and tolerance to functional groups. We also demonstrated that the efficient HUM formation is uniquely driven by both kinetic and thermodynamic effects, which, more interestingly, share a common contributor: the steric *N*-substituent. Bulky groups are playing multiple roles to achieve high-efficient macrocycle synthesis here:

first of all, it activates the covalent urea bond to provide reversible feature and enable systems to find their minimum energy state; secondly, it promotes the formation speed of the macrocycles by conformation lock, which increases the effective molarity of local reactants and the probability of the ring-closing steps; thirdly, it reduces the energy and promote the dominance of the macrocycle system through weak interactions with the ring pocket. The contribution of 'kinetic' and 'thermodynamic' effect to the macrocycle synthesis may vary, depending on variables in each specific macrocycle structure, such as ring size, monomer length/angle, etc. Even more uniquely, the bulky group can be instantly removed with the catalysis of acid, leaving stable macrocycles without further concerns of structural changes.

Apart from introducing an efficient way to obtain a class of functional macrocyclic molecules, this research underscored the power of harnessing bulky group to facilitate highly efficient preparation of macrocycles both kinetically and thermodynamically. We envision this strategy could also be applied to the preparation of more complex architectures and to other DCC systems.

Here we report a simple, high-yielding, and robust method for constructing urea macrocycles with dynamic HUB chemistry and acid-assisted de-*tert*-butylation. HUMs can be formed in nearly quantitative yields under mild conditions with high concentrations in a relatively short timescale. The unusual high efficiency was attributed to the 'self-correction' property enabled by the dynamic bond as well as the 'cis-urea preference' and weak stabilization effect mediated by the bulky group. In addition, the bulky group can be efficiently removed by acid to stabilize the macrocycle structure, which in turn showed higher propensity for self-assembly, strong binding affinity for anions and potential as an antimicrobial peptide surrogate for killing of drug-resistant microbials. We envision that this 'bulky group effect' may not be limited to HUB but exist in other DCC systems as well. Given the exceptional simplicity, high yield synthesis, easy functionalization, flexibility in substrate selection and hydrogen bonding capacity of the product, we expect this chemistry to become a great platform for the study of macrocycle applications in areas, such as self-assembly, selective binding, molecular recognition, drug discovery, catalysis and rotaxane, etc.

## Methods

**Full experimental details and characterization of compounds can be found in the Supplementary Information**. Typical hindered diamine synthesis: The corresponding dihalides was dissolved in 10 mL DMF. *Tert*-butylamine (6 eq) and $K_2CO_3$ (1 eq) were added to the solution. The suspension was stirred at room temperature for about 24 h and the reaction was monitored by TLC. After completion, the reaction was quenched with 20 mL water, and then extracted with DCM (30 mL × 3). The organic layer was combined, washed twice with brine and then dried with anhydrous $Na_2SO_4$. Then solvent was removed and crude product was purified by flash column chromatography.

Macrocycle library generation: The diisocyanate (0.1 mmol) and diamine (0.1 mmol) were mixed directly in $CDCl_3$ (2 mL) and then incubated at 60 °C. The reactions were monitored by $^1H$ NMR until the thermodynamic equilibrium had been reached (evidenced by the lack of change in $^1H$ NMR). Final products were further confirmed by $^{13}C$ NMR and MALDI-TOF. The yields were calculated by the integration from $^1H$ NMR spectra.

Macrocycle formation kinetics: The diisocyanate (0.1 mmol in 1 mL $CDCl_3$) and diamine (0.1 mmol in 0.5 mL $CDCl_3$) were quickly mixed, rinsed with 0.5 mL $CDCl_3$ and then subjected to NMR immediately. Then the mixtures were kept at 60 °C and NMR spectra were taken at various intervals. The yields were calculated by the integration from $^1H$ NMR spectra.

Calculations: Quantum chemistry calculations using the Gaussian09 package were performed to determine the relative structural interaction energies of different model compounds (*cis/trans* and different conformers). To achieve high accuracy, calculations were performed with DFT and Møller–Plesset second order perturbation (MP2) theory. For the DFT calculations, PBE functional at generalized gradient approximations level and Becke's three parameter hybrid exchange functional and Lee–Yang–Parr correlation functional at hybrid level were selected. 6-31/G(d,p) basis set was used for both DFT functionals and aug-cc-pVDZ basis set was used for MP2. All calculations were performed at gas phase.

## Data availability

The data supporting this study are provided in the Supplementary Information and are also available from the authors upon reasonable request. All single-crystal data of **HUM1**, **HUM3**, **[N4A2]2** and **MC5** have been deposited in the Cambridge Crystallographic Data Centre (CCDC) and can be downloaded free of charge from http://www.ccdc.cam.ac.uk/data_request/cif. The accession numbers are CCDC: 1959553, 1959555, 1959556 and 1959562, respectively.

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

## Acknowledgements

This work is supported by United States National Science Foundation (NSF CHE 17-09820) and American Chemical Society Petroleum Research Fund (58671-ND7). We thank R. Wang for the helpful discussion, M. Kang for the WAXS characterization and J. Lai for the assistance with computation of the binding constants.

## Author contributions

Y.Y. and H.Y. contributed equally to this work. Y.Y., H.Y. and J.C. conceived the idea of the project. Y.Y. and H.Y. performed the experimental work. Z.L., J.W., A.F. and Y.Z. performed the simulations. Y.C. helped with some synthesis. D.G. performed the XRD studies and analyzed the data. B.L. and Q.C. took the SEM image. Y.Y., H.Y. and J.C. wrote the manuscript with contributions from all authors. All authors discussed the results and commented on the manuscript.

## Competing interests

The authors declare no competing interests.
