## [Peer Review File · Nature Communications]

REVIEWER COMMENTS

Reviewer #1 (Remarks to the Author):

This is a clever and exciting method for catalyzing the synthesis of urea macrocycles using dynamic hindered urea bond. The simple one pot method works in very high yield at high concentration of equal molar ratio of diisocyanate and hindered diamines. This important application of dynamic covalent chemistry under kinetic control has been beautifully documented to make a library of urea macrocycles. This innovation is going to open the area of urea macrocycles and greatly enhance investigation of their utility in areas of medicinal chemistry, supramolecular chemistry, selective binding, rotaxane formation and as porous organic solids for catalysis. The 'bulky group effect' may be applicable to other dynamic covalent syntheses as well. The supporting materials are complete and readily accessible.

Reviewer #2 (Remarks to the Author):

This paper by Cheng and colleagues describes an extension of their past work on dynamic urea bonds formed between isocyanates and bulky substituted amines, here showing the formation of discrete macrocyclic species. This piece was a pleasure to read, exceptionally well rationalized, with rigorous data collection and molecular characterization. There are many exciting possibilities to use this technique, and I look forward to the works which will surely follow. This reviewer has only a few comments to consider in further strengthening the work.

1- With regard to the "pocket effect" which the authors attribute, in part, to driving the formation of discrete species, it would be interesting to further probe concentration in order to determine if the realization of discrete macrocyclic species holds under conditions more dilute than the range presently probed here (1mM-500mM).

2- The assemblies shown in Figure 5d are interesting, and it would be appreciated if some additional characterization of these could be provided. For instance, FTIR, circular dichroism, polarized light microscopy, thermal stability, etc. To be clear, this present data appears somewhat tangential to the objectives of the current paper and it may instead be desirable for the authors to publish a followup piece on the self-assembly of these macrocycles.

Reviewer #3 (Remarks to the Author):

In this manuscript, the authors reported a quite effective synthetic method for urea macrocycles. The strategy is to use dynamic hindered urea bond. Diisocyanate and hindered diamine with bulky t-butyl groups can react under high concentrations to afford the macrocycles in almost quantitative yields. Further treatment with acid can cleave the t-butyl group and gives rise to urea macrocycles. The role of the bulky N-t-butyl group was found to play an important role in the macrocyclization through kinetic control and thermodynamic stabilization. The method is rather general and can be applied to the synthesis of a variety of urea macrocycles. Moreover, some of the urea macrocycles

were demonstrated to self-assemble into nanometer fibers, to be potent receptors for anions, or to show antimicrobial activity. This is an interesting research which is inspiring for macrocycle synthesis. Therefore, I would like to support its publication in Nat. Commun. after the authors address the following issues:

- a) "Quantitative synthesis" in the title is not consistent with the reality: the yields are shown to be > 95% or lower. This should be expressed more accurately. In fact, the overall yields of the two steps (macrocyclization and acid-assisted cleavage of t-butyl group) should be considered here.
- b) With respect to the dimeric complex in the crystal structure, this may not exist in solution. Crystal structures are determined by two factors: crystal packing and noncovalent interactions. Considering the nature of the groups, the interactions between the t-butyl group and the cavity are C-H...pi and dispersion, and are thus very weak. I do not see the possibility to maintain the dimer in solution. If the authors want to confirm this, it is better to find some experimental evidences besides computational results, for example, through determining the dimeric association constant in nonpolar solvents. Furthermore, the thermodynamic role of the t-butyl group should be re-analyzed by considering new experimental results.
- c) There are too many abbreviations which hinder smooth reading of the manuscript. Please judiciously use abbreviations only when they are widely accepted and easily understood.
- d) There are some typos, for example, "to some extend".

Response to the reviewers' comments on NCOMMS-20-40954

Response to reviewer #1:

This is a clever and exciting method for catalyzing the synthesis of urea macrocycles using dynamic hindered urea bond. The simple one pot method works in very high yield at high concentration of equal molar ratio of diisocyanate and hindered diamines. This important application of dynamic covalent chemistry under kinetic control has been beautifully documented to make a library of urea macrocycles. This innovation is going to open the area of urea macrocycles and greatly enhance investigation of their utility in areas of medicinal chemistry, supramolecular chemistry, selective binding, rotaxane formation and as porous organic solids for catalysis. The 'bulky group effect' may be applicable to other dynamic covalent syntheses as well. The supporting materials are complete and readily accessible.

We thank the reviewer for the kind remarks!

Response to reviewer #2:

This paper by Cheng and colleagues describes an extension of their past work on dynamic urea bonds formed between isocyanates and bulky substituted amines, here showing the formation of discrete macrocyclic species. This piece was a pleasure to read, exceptionally well rationalized, with rigorous data collection and molecular characterization. There are many exciting possibilities to use this technique, and I look forward to the works which will surely follow. This reviewer has only a few comments to consider in further strengthening the work.

1- With regard to the "pocket effect" which the authors attribute, in part, to driving the formation of discrete species, it would be interesting to further probe concentration in order to determine if the realization of discrete macrocyclic species holds under conditions more dilute than the range presently probed here (1mM-500mM).

Thanks for the great question. As the reviewer suggested, we tried 0.5 mM and 0.2 mM, both showed similar results, with discrete macrocyclic species obtained in near quantitative yields. For further dilution of reactants, impurities from the solvent (for example, chloroform usually contains ethanol as stabilizer) may interfere the reaction. Under these conditions, the preference for the macrocycles formation is likely to be more “kinetic-controlled” (less competition from intermolecular reaction) and less “thermodynamic controlled” (less tendency to self-associate). One of the goals for our future studies is to de-couple the two effects and understand their contributions in different systems.

2- The assemblies shown in Figure 5d are interesting, and it would be appreciated if some additional characterization of these could be provided. For instance, FTIR, circular dichroism, polarized light microscopy, thermal stability, etc. To be clear, this present data appears somewhat tangential to the objectives of the current paper and it may instead be desirable for the authors to publish a follow up piece on the self-assembly of these macrocycles.

Thanks for the suggestions. We agree the suggested new studies are important to understand the self-assembly behaviors in more details. But since the focus of this paper is on the chemistry aspect. We only touched the potential application slightly and have planned to conduct thorough and comprehensive study of these structure characterization in our future work.

Response to reviewer #3:

In this manuscript, the authors reported a quite effective synthetic method for urea macrocycles. The strategy is to use dynamic hindered urea bond. Diisocyanate and hindered diamine with bulky t-butyl groups can react under high concentrations to afford the macrocycles in almost quantitative yields. Further treatment with acid can cleave the t-butyl group and gives rise to urea macrocycles. The role of the bulky N-t-butyl group was found to play an important role in the macrocyclization through kinetic control and thermodynamic stabilization. The method is rather general and can be applied to the

synthesis of a variety of urea macrocycles. Moreover, some of the urea macrocycles were demonstrated to self-assemble into nanometer fibers, to be potent receptors for anions, or to show antimicrobial activity. This is an interesting research which is inspiring for macrocycle synthesis. Therefore, I would like to support its publication in Nat. Commun. after the authors address the following issues:

a) “Quantitative synthesis” in the title is not consistent with the reality: the yields are shown to be > 95% or lower. This should be expressed more accurately. In fact, the overall yields of the two steps (macrocyclization and acid-assisted cleavage of t-butyl group) should be considered here.

Thanks for the suggestions. We agree with the suggestion from the reviewer, and changed to “near quantitative synthesis” in our title accordingly. In most of our examples, side reactions are negligible according to the data but we agree no system is perfect.

To clarify, when reporting the yields of >95% for those near quantitative reactions, the consideration is more based on the error of 5% from the NMR spectral processing software.

For the macrocyclization step, the yields were reported in table 1. For the de-tert-butylation step, the synthetic yield is also quantitative for all reactions we tried. The separation yield is a little less than quantitative since it involves precipitation of the sample to remove excessive amount of acid, which will cause loss of the samples. If we consider the overall yields of the two steps combined, the yield will still be near quantitative for most of our examples.

b) With respect to the dimeric complex in the crystal structure, this may not exist in solution. Crystal structures are determined by two factors: crystal packing and noncovalent interactions. Considering the nature of the groups, the interactions between the t-butyl group and the cavity are C-H...pi and dispersion, and are thus very weak. I do not see the possibility to maintain the dimer in solution. If the authors want to confirm this, it is better to find some experimental evidences besides computational results, for example, through determining the dimeric association constant in nonpolar solvents. Furthermore, the thermodynamic role of the t-butyl group should be re-analyzed by considering new experimental results.

We thank the reviewer for the insightful suggestion and sorry for the misunderstanding caused by our claims. We agree that the interactions between the tert-butyl group and the cavity are very weak. We are not expecting a persistent “dimeric complex” in solution, thus it might cause confusion to say “host-guest interaction” or “template effect” in our manuscript.

As the reviewer suggested, we tested the dimeric association constant of HUM1 in chloroform and the K_a was determined to be around 5.5 M^{-1} (data added in the manuscript). Although very weak, the association do exist in solution. Apart from that, we also have some other evidence as shown in the manuscript. In the concentration dependent NMR, the peak of the tert-butyl group gradually shifted upfield with increasing concentration, showing the tert-butyl group is being shielded. The NOE spectrum also demonstrated interaction between the macrocycle and a model compound. To further support the hypothesis, we conducted a series of computational studies which can well support our idea. Based on our experimental results and computational studies, we think the weak interactions might play a role. There are a few reports from literatures showing similarly weak interactions impacting the distribution of final products or reaction kinetics. In one example, Jiao etc. reported the quantitative formation of an imine cage driven by weak C-H... π interactions, without which the yield is significantly lower (Jiao *et al.*, *Angew. Chem.* **2017**, 129, 14737–14742). In our another just accepted work, although the interactions between crown ether and amines are also very weak and can hardly be accurately determined, the addition of crown ether can greatly enhance the ring opening kinetics of amines.

We thank the reviewer again for the suggestions. We have removed the misleading wordings like “host-guest interaction” and “template effect” in our manuscript. Instead, we have emphasized that they are “weak interactions” which possibly contributed the high yields of macrocycle formation. Changes were highlighted in yellow.

Figure. Determination of dimeric association constant of HUM1 in CDCl_3 .

c) There are too many abbreviations which hinder smooth reading of the manuscript. Please judiciously use abbreviations only when they are widely accepted and easily understood.

Thanks for the suggestion from the reviewer. We did not mean to abuse the abbreviations but agreed that too many of those hamper efficient scientific communications. We already removed some unnecessary abbreviations in our revised manuscript. More specifically, HB (hydrogen bonding), MW (molecular weight), AMP (antimicrobial peptide) have been replaced with their corresponding full descriptions. A few abbreviations were kept because they were either widely accepted in dynamic systems (DCC for dynamic covalent chemistry), or occurred frequently in our manuscript (HUM for hindered urea macrocycles). Changes were highlighted in blue.

d) There are some typos, for example, “to some extend”.

Thanks for the correction. We have done a thorough check of our manuscript again and tried our best to get rid of any typos or grammatic issues.

REVIEWERS' COMMENTS

Reviewer #2 (Remarks to the Author):

The authors have addressed my concerns and I believe the paper is now suitable for publication in Nature Communications. I look forward to exploring this concept further once the paper is out, and thank the authors for this detailed and insightful work.

Reviewer #3 (Remarks to the Author):

No further comments. The authors have well addressed my concerns. I fully support its publication.